# Prolyl Oligopeptidase Regulates Dopamine Transporter Oligomerization and Phosphorylation in a PKC- and ERK-Independent Manner

**DOI:** 10.3390/ijms22041777

**Published:** 2021-02-10

**Authors:** Ulrika H. Julku, Maria Jäntti, Reinis Svarcbahs, Timo T. Myöhänen

**Affiliations:** 1Division of Pharmacology and Pharmacotherapy/Drug Research Program, Faculty of Pharmacy, University of Helsinki, Viikinkaari 5E (P.O. Box 56), FI-00014 Helsinki, Finland; ulrika.julku@pubcare.uu.se (U.H.J.); maria.jantti@helsinki.fi (M.J.); reinissv@outlook.com (R.S.); 2Integrative Physiology and Pharmacology Unit/Institute of Biomedicine, Faculty of Medicine, University of Turku, FI-20520 Turku, Finland

**Keywords:** dopamine, serine protease, alpha-synuclein, extracellular signal-regulated kinase, protein kinase C

## Abstract

Prolyl oligopeptidase (PREP) is a serine protease that binds to alpha-synuclein (aSyn) and induces its aggregation. PREP inhibitors have been shown to have beneficial effects in Parkinson’s disease models by enhancing the clearance of aSyn aggregates and modulating striatal dopamine. Additionally, we have shown that PREP regulates phosphorylation and internalization of dopamine transporter (DAT) in mice. In this study, we clarified the mechanism behind this by using HEK-293 and PREP knock-out HEK-293 cells with DAT transfection. We tested the effects of PREP, PREP inhibition, and alpha-synuclein on PREP-related DAT regulation by using Western blot analysis and a dopamine uptake assay, and characterized the impact of PREP on protein kinase C (PKC) and extracellular signal-regulated kinase (ERK) by using PKC assay and Western blot, respectively, as these kinases regulate DAT phosphorylation. Our results confirmed our previous findings that a lack of PREP can increase phosphorylation and internalization of DAT and decrease uptake of dopamine. PREP inhibition had a variable impact on phosphorylation of ERK dependent on the metabolic state of cells, but did not have an effect on phosphorylation or function of DAT. PREP modifications did not affect PKC activity either. Additionally, a lack of PREP elevated a DAT oligomerization that is associated with intracellular trafficking of DAT. Our results suggest that PREP-mediated phosphorylation, oligomerization, and internalization of DAT is not dependent on PKC or ERK.

## 1. Introduction

Dopamine (DA) transporter (DAT) is a plasma membrane protein that acts as the main regulator of dopaminergic signaling by transporting extracellular DA into presynaptic neurons. Dopaminergic neurotransmission is impaired in many diseases, such as Parkinson’s disease, depression, bipolar disorder, schizophrenia, attention-deficit/hyperactivity disorder, and addictions [1,2]. Phosphorylation of DAT regulates DA transport capacity as phosphorylated DAT is internalized, decreasing DAT on the plasma membrane and uptake of DA [3,4]. DAT function is regulated by several kinases such as protein kinase C (PKC), calcium-calmodulin-dependent protein kinase II (CaMKII), extracellular signal-regulated kinase (ERK), and protein kinase A (PKA) [3,5,6,7,8,9]. ERK is a negative regulator for DA signaling, but DA does not activate ERK [10,11,12]. Activation of ERK by phosphorylation causes phosphorylation of mitogen-activated protein kinase/SRC homology 3 (MAPK/SH3) domain of DAT-inducing upregulation of DAT surface level and transport capacity, which leads to reduced extracellular DA and dopaminergic neurotransmission. Active PKC phosphorylates the PKC domain of DAT, leading to increased DAT phosphorylation, enhanced DAT internalization, and reduced DA uptake [5,13,14], indicating PKC as positive regulator of DA neurotransmission. Interestingly, DAT is also expressed as oligomer particularly in cytosol, and this has been connected with intracellular trafficking (for review, see [15]). It is not clear if oligomeric forms of DAT are functional in the plasma membrane, but it appears that at least stimulants such as amphetamine impact DAT oligomerization [15].

Alpha-synuclein (aSyn), the main component of Lewy bodies and the key player in neurotoxicity in Parkinson´s disease [16], is also shown to regulate DAT function by stabilizing DAT on the plasma membrane [17,18]. Increased DAT on the plasma membrane can lead to elevated intracellular levels of DA and oxidative stress; furthermore, increased DA induces aSyn oligomerization and prevents aSyn fibrillization [19,20]. Additionally, aSyn binds to ERK2 in vitro [21], and overexpression of aSyn has been shown to induce ERK phosphorylation in cells [22,23]. However, there have also been controversial results of aSyn overexpression decreasing ERK phosphorylation and impairing PKC activity, but not affecting expression levels of ERK [24]. It has been shown that even a short incubation with aSyn can activate ERK1/2 in cells, indicating receptor-mediated activation of the MAPK signaling pathways [25]. aSyn-induced changes in ERK phosphorylation have also been found clinically, as phosphorylated ERK is elevated in the cytoplasm and mitochondria of midbrain dopaminergic neurons in the human brain in Parkinson´s disease and diffuse Lewy body disease [26,27].

We have shown that a serine protease, prolyl oligopeptidase (PREP), regulates DAT phosphorylation and internalization in the mouse striatum [28]. Additionally, PREP binds to aSyn and increases its aggregation in vitro and in vivo [29,30,31], and PREP inhibitors can block the aggregation and enhance the clearance of aggregates in aSyn overexpressing cells and in aSyn transgenic mice [32]. PREP is widely distributed in different tissues, but there is high activity in the brain, especially in GABAergic, cholinergic, and dopaminergic neurons [33,34], and a study by Jalkanen et al. [35] showed that PREP inhibitors can also reduce extracellular DA in the rat brain. As another link to the interplay among DAT, aSyn, and PREP, PREP is involved in regulation of ERK function [36,37]. PREP inhibitors JTP-4819 and KYP-2047 decrease phosphorylation of ERK1/2, especially ERK2 in SH-SY5Y cells. However, regulation of ERK phosphorylation by PREP may also have an indirect mechanism, because both treatment with purified PREP and PREP pretreated serum increase phosphorylation of ERK in cells [37].

Based on PREP’s interaction with ERK and DAT, we hypothesized that PREP could regulate DAT phosphorylation via ERK. In this study, we aimed to characterize PREP as a regulator of DAT and tried to find the mechanism explaining the previous findings with PREP, DAT, and ERK by using HEK-293 cells transfected with DAT, Western blot (WB), and a dopamine uptake assay in the presence and absence of PREP. Additionally, we characterized the impact of aSyn and PREP inhibition on DAT, and evaluated if the PREP-mediated effect is based on PKC or ERK regulation by using kinase assay and Western blot.

## 2. Results

The aim of this study was to characterize the mechanisms of PREP on regulating DAT function. We studied the effect of overexpression, inhibition, or absence of PREP on DAT and ERK by WB, a DA uptake assay, and a PKC activity assay. Additionally, we investigated whether PREP can change aSyn-mediated regulation of DAT function.

### 2.1. The Effect of PREP and α-Synuclein on Phosphorylated and Oligomeric DAT, and Phosphorylation of ERK

The effect of PREP and aSyn on phosphorylation of DAT and ERK was studied in Wt and stable PREPko HEK-293 cells. Interestingly, our DAT transfection-based model gave rise to DAT oligomers in cells (Appendix A). We also assessed the effect of PREP and aSyn on DAT oligomerization. To characterize the role of PREP, the oligomeric DAT (oDAT), phosphorylated DAT (pDAT), ERK, and phosphorylated ERK (pERK) were measured by WB in cells that were transfected with DAT and active or inactive PREP (S554A-PREP). The role of PREP in aSyn-regulated phosphorylation of DAT and ERK was studied in DAT and aSyn transfected cells in the absence and presence of PREP.

Overexpression of PREP or S554A-PREP in HEK-293 cells did not have a statistically significant effect on pDAT (F5, 35 = 1.105, *p* = 0.378, one-way ANOVA, Figure 1A), total ERK (F5,35 = 1.110, *p* = 0.376, one-way ANOVA, Appendix A), or pERK/ERK ratio (F5,35 = 0.347, *p* = 0.880, one-way ANOVA, Figure 1C, Appendix A), but inactive PREP-S554A increased oDAT (F5,29 = 2.714, *p* = 0.041, one-way ANOVA with Tukey´s post-hoc test, Figure 1B). However, active PREP did not have a significant effect on oDAT (F5,29 = 2.714, *p* = 0.622, one-way ANOVA, Tukey´s post-hoc test, Figure 1B). Restoring PREP activity to PREPko HEK-293 cells (F5,29 = 2.714, *p* = 0.997, one-way ANOVA, Tukey´s post-hoc test, Figure 1B) or transfection with inactive S554A-PREP (F5,29 = 2.714, *p* = 0.943, one-way ANOVA, Tukey´s post-hoc test) compared to mock-transfection did not have a significant effect on oDAT in PREPko cells. Two-way ANOVA revealed elevated pDAT in PREPko HEK-293 cells compared to Wt HEK-293 cells (F1,35 = 4.713, *p* = 0.038, two-way ANOVA, Figure 1A). Cell line also changed the effect of transfection on oligomeric DAT (cell line*transfection interaction, F2,29 = 4.469, *p* = 0.022, two-way ANOVA, Figure 1B), and there was a similar trend for total ERK (cell line*transfection interaction: F2,35 = 2.566, *p* = 0.094, two-way ANOVA, Appendix A).

Although some changes were seen, the overexpression of aSyn, or aSyn and PREP did not have a statistically significant effect on pDAT (F5,35 = 1.232, *p* = 0.319, one-way ANOVA, Figure 2A), oDAT (F5,35 = 1.489, *p* = 0.223, one-way ANOVA, Figure 2B), or pERK/ERK ratio (F5,35 = 0.834, *p* = 0.536, one-way ANOVA, Figure 2C; see Appendix A-D for ERK and pERK) in Wt and PREPko HEK-293 cells (Figure 2D). Since we have shown in several studies that PREP inhibition decreases aSyn aggregation, we tested here if a PREP inhibitor, KYP-2047, has an effect on DAT or ERK in absence of aSyn. Two-way ANOVA revealed that KYP-2047 treatment on aSyn or aSyn+PREP transfected Wt HEK-293 cells increased phosphorylation of ERK (F1,47 = 18.584, *p* = 0.00096, two-way ANOVA, Figure 2G,H; see Appendix A for ERK and pERK), but the difference between the single groups was not statistically significant due to high variation. Moreover, KYP-2047 treatment or transfection with aSyn or a combination of aSyn and PREP did not have an effect on pDAT (F5,47 = 1.329, *p* = 0.271, one-way ANOVA, Figure 2E), oDAT (F5,47 = 2.319, *p* = 0.060 one-way ANOVA, Figure 2F), or ERK (F5,23 = 0.845, *p* = 0.536, one-way ANOVA, Appendix A).

### 2.2. The Effect of PREP Inhibitor KYP-2047 on DAT and Phosphorylation of ERK

To assess the impact of PREP inhibition by KYP-2047 on DAT and ERK, DAT-transfected HEK-293 cells were incubated with 1 µM KYP-2047 for 30 min with and without serum starvation. A PKC activator, PMA, was used as control with starved cells and a PKC inhibitor, Gö-6983, was used on non-starved cells as a reference for PKC modification. PREP inhibitor KYP-2047 did not have a significant influence on pDAT (Figure 3D; F3,23 = 3.840, *p* = 0.999, one-way ANOVA, Tukey´s post-hoc test, Figure 3A), oDAT (F3,23 = 1.246, *p* = 1.000, one-way ANOVA, Tukey´s post-hoc test, Figure 3B), or pERK/ERK ratio (F3,23 = 12.370, *p* = 1.000, one-way ANOVA, Tukey´s post-hoc test, Figure 3C; see Appendix A for ERK and pERK) in starved cells or without starvation (pDAT: F3,23 = 2.293, *p* = 0.893, Figure 3E; oDAT: F3,23 = 0.859, *p* = 0.763, Figure 3F; pERK/ERK: F3,23 = 3.373, *p* = 0.997, Figure 3G, one-way ANOVA, Tukey´s post-hoc test; see Appendix A for ERK and pERK). The positive control PMA increased phosphorylation of ERK (F3,23 = 12.370, *p* = 0.003, one-way ANOVA, Tukey´s post-hoc test, Figure 3C) and elevated phosphorylated DAT (F3,23 = 3.840, *p* = 0.120, one-way ANOVA, Tukey´s post-hoc test, Figure 3A) as expected, but this was not statistically significant. PMA did not cause alterations in oligomeric DAT (F3,23 = 1.246, *p* = 0.969, one-way ANOVA, Tukey´s post-hoc test, Figure 3B). A combination of KYP-2047 and PMA increased pDAT slightly more than PMA alone (F3,23 = 3.840, *p* = 0.078, one-way ANOVA, Tukey´s post-hoc test, Figure 3A), and this was also seen in oDAT (Figure 3B), but the differences between the treatments were not significant. Additionally, the pERK/ERK ratio was increased after KYP-2047 and PMA treatment (F3,23 = 12.370, *p* = 0.001, one-way ANOVA, Tukey´s post-hoc test, Figure 3C), but there was no difference on PMA alone.

In non-starved cells, PKC inhibitor Gö-6983 did not affect phosphorylation of DAT (F3,23 = 2.293, *p* = 0.382, one-way ANOVA, Tukey´s post-hoc test, Figure 3E), but in combination with KYP-2047, seemed to potentiate the effect of Gö-6983 on pDAT (F3,23 = 2.293, *p* = 0.099 one-way ANOVA, Tukey´s post-hoc test, Figure 3E). Gö-9683 also decreased the pERK/ERK ratio, but this was not statistically significant (F3,23 = 3.373, *p* = 0.092, one-way ANOVA, Tukey´s post-hoc test, Figure 3G). KYP-2047, Gö-6983, or their combination did not have an effect on oDAT, but in contrast to PREPko and starved cells, KYP-2047 decreased oDAT levels (KYP-2047: F3,22 = 0.859, *p* = 0.763; Gö-6983: F3,22 = 0.859, *p* = 0.936; KYP-2047 + Gö-6983: F3,22 = 0.859, *p* = 0.967, one-way ANOVA, Tukey´s post-hoc test, Figure 3F). KYP, PMA, or Gö-6983 did not have an effect on expression of ERK (PMA + KYP: F3,15 = 0.596, *p* = 0.629; Gö-6983 + KYP: F3,15 = 0.745, *p* = 0.546, one-way ANOVA, Appendix A).

### 2.3. The Effect of PREP Inhibitor KYP-2047 on PKC Activity

PKC can directly phosphorylate both ERK and DAT, so we wanted to see if PREP inhibition had any effect on PKC activity (Figure 3H). We used PMA as a positive control and PKC inhibitor Gö-6983 as a negative control. As expected, PMA induced an increase in PKC activity (although this was not statistically significant, F5,15 = 10.63, *p* = 0.0736, one-way ANOVA, Tukey´s post-hoc test, Figure 3H), whereas Gö-6983 alone did not have much effect. Gö-6983 abolished the effect of PMA when they were used in combination, verifying that the PMA-induced signal was indeed PKC activity. When used alone, KYP-2047 did not affect the activity of PKC, but surprisingly, it potentiated the effect of PMA when these two were combined (F5,15 = 10.63, *p* = 0.0182, one-way ANOVA, Tukey´s post-hoc test, Figure 3H).

### 2.4. PREP and DAT as Regulators of Dopamine Uptake

The function of DAT was studied by using a 3H-DA uptake assay in Wt and PREPko HEK-293 cells transfected with DAT and active or inactive PREP. The impact of aSyn on DAT functionality was studied by using combinations of DAT transfection with aSyn +/− PREP. The effect of PREP inhibition on DA uptake was studied in DAT-transfected cells that were treated with PREP inhibitor KYP-2047 alone or in combination with PKC activator or inhibitor. Overexpression of PREP or inactive PREP did not affect DA uptake in Wt or PREPko HEK-293 cells (Figure 4A). However, DA uptake was decreased in aSyn and DAT-transfected PREPko cells (F5,52 = 3.957, *p* = 0.037, one-way ANOVA, Tukey´s post-hoc test, Figure 4B), and further decreased when PREP was added to this combination (F5,52 = 3.957, *p* = 0.002, one-way ANOVA, Tukey´s post-hoc test, Figure 4B) when compared to Wt HEK-293 cells transfected with DAT and mock. A similar effect was not seen in Wt HEK-293 cells. Interestingly, DA uptake was decreased in mock and DAT-transfected PREPko cells (F5,52 = 3.957, *p* = 0.086, one-way ANOVA, Tukey´s post-hoc test, Figure 4B). Two-way ANOVA revealed a statistically significant difference between Wt and PREPko cells (cell line effect; F1,52 = 12.442, *p* = 0.000949, two-way ANOVA, Figure 4B) and between mock and DAT and PREP, aSyn and DAT-transfected cells (treatment effect; F2,52 = 3.442, *p* = 0.037, two-way ANOVA, Tukey´s post-hoc test, Figure 4B). The difference in DA uptake between DAT- and mock-transfected Wt HEK-293 and PREPko cells became visible only in the setup in which aSyn or aSyn+PREP was transfected on cells (Figure 4B), but not in DAT+PREP or DAT+PREP-S554A transfected cells (Figure 4A). This was unexpected, and could be due the different amount of DAT and mock plasmids in these groups, as the amount of plasmids was adjusted for three different constructs in Figure 4B.

KYP-2047 did not have a statistically significant effect on DA uptake in starved Wt HEK-293 cells (F3,32 = 3.052, *p* = 0.6432, one-way ANOVA, Tukey´s post-hoc test, Figure 4C) or without starvation (F3,31 = 0.517, *p* = 0.4549, one-way ANOVA, Tukey´s post-hoc test, Figure 4D). A combination of PMA and KYP-2047 significantly decreased uptake of DA (F3,32 = 3.052, *p* = 0.0256, one-way ANOVA, Figure 4C). PMA alone decreased DAT uptake, but did not reach significance (F3,32 = 3.052, *p* = 0.4130, one-way ANOVA, Figure 4C). PKC inhibitor Gö-6983 did not have a statistically significant effect on DA uptake alone (F3,31 = 0.517, *p* = 0.1801, one-way ANOVA, Tukey´s post-hoc test, Figure 4D), or in combination with KYP-2047 (F3,31 = 0., *p* = 0.1825, one-way ANOVA, Tukey´s post-hoc test, Figure 4D). Generally, the changes in the DA uptake assay were minor, possibly due to the fact that most of the DAT was expressed in oligomeric form.

## 3. Discussion

We showed earlier that PREP participates in the regulation of DAT phosphorylation in mouse striatum, and a lack of PREP leads to increased DAT phosphorylation and elevated extracellular DA [28]. Additionally, long-term PREP inhibitor treatment for aSyn transgenic mouse significantly elevated striatal dopamine levels [29]. In this study, we investigated the mechanisms in the cellular model to explain the previous in vivo results. Since PREP has been shown to regulate the phosphorylation of ERK [36,37], and PKC-mediated ERK phosphorylation is one of the main regulators for phosphorylation of DAT [10,11], our hypothesis was that PREP could regulate DAT via ERK. However, the results of this study revealed that ERK does not participate in PREP-mediated regulation of DAT. PREP seemed to regulate phosphorylation of DAT and ERK in cells, but the mechanism appeared to be independent from PKC, and changes in ERK phosphorylation did not correlate with DAT phosphorylation levels. Our study also showed that PREP participates in the regulation of DAT oligomerization, and lack of PREP elevates the levels of DAT oligomers that may contribute to elevated phosphorylation of DAT.

ERK has been shown to phosphorylate the Thr53 site in the MAPK/SH3 domain, and this phosphorylation stabilizes DAT on the cellular membrane and increases uptake of DA [38,39,40,41]. We observed an elevated phosphorylation of DAT in PREPko cells compared to Wt HEK-293 cells in WB assay, but a slightly decreased uptake of 3H-DA in the DA uptake assay. A similar phenomenon was found in PREPko mice, as pDAT was increased in the striatal tissue, and extracellular DA was elevated in the striatum in PREPko mice compared to Wt littermates in our previous study [28]. Interestingly, elevated pDAT levels in PREPko cells was accompanied by elevated oDAT levels. DAT oligomerization is connected with DAT transport from ER-golgi to plasma membrane [15], and phosphorylation of DAT at Thr53 is also associated with ERK-mediated tonic transport of DAT (for review, see [41]). Therefore, elevated pDAT and DAT oligomerization in PREPko cells is pointing to elevated transport of DAT from membrane to cytosol. PREP forms interactions with the microtubule network, and it has been suggested that PREP could be involved with protein-transport functions [42]. Hence, it is possible that a lack of PREP can lead to impaired intracellular transport of DAT.

Overexpression of PREP or inactive S554A-PREP did not change the pDAT or pERK/ERK ratio in Wt cells, although in our previous study overexpression of PREP lowered pDAT in the striatum of mice [28]. Although a lack of PREP elevated DAT oligomerization, overexpression of PREP or S554A-PREP elevated DAT oligomerization in cells as well, which is controversial. Overexpression in the cells may lead to rapid adaptation and a lack of response, and additionally, lipofectamine transfection may cause stress to the cells and could explain the difference. Similar controversial results were seen when PREP was restored to PREPko cells. Although restoring PREP to PREPko mice did not have any effects on DAT phosphorylation and function [28], restoring PREP or S554A-PREP in PREPko cells further elevated pDAT levels and active PREP, as well as oligomeric DAT. It is likely that lack of PREP has led to an adaptive mechanism, and restoring PREP cannot overcome these as seen in mice. Another explanation could be, as we have recently shown, that PREP negatively regulates protein phosphatase 2A (PP2A) [43]. PP2A is one of the phosphatases that dephosphorylates DAT [44], and our previous study showed that restoring PREP or S554A-PREP in PREPko cells inhibits PP2A effectively [43], which may well be seen in PREPko cells as elevated DAT phosphorylation. This can then lead to elevated DAT oligomerization if DAT is directed from the plasma membrane to the cytosol. However, inactive PREP did not elevate DAT oligomers, suggesting that PREP activity is required for certain actions in the DAT regulation pathway.

Treatment with purified PREP or with a PREP inhibitor was shown to alter phosphorylation of ERK in mouse macrophage cells in an earlier study [37], but we did not observe statistically significant changes in phosphorylation of ERK in PREP-overexpressing HEK-293 cells, but in PREPko cells, PREP and inactive PREP elevated the pERK/ERK ratio. In normal conditions, PREPko cells had slightly lower ERK phosphorylation levels, and PREP inhibition did not alter ERK phosphorylation. Interestingly, when HEK-293 cells were transfected with combinations of DAT, aSyn, and PREP, KYP-2047 elevated the ERK phosphorylation significantly, while the opposite effect was observed in mouse macrophages in which PREP inhibition significantly decreased ERK phosphorylation [37]. This indicates that PREP participates in regulating ERK phosphorylation, and this regulation is dependent on the cellular metabolism state and possibly on cellular stress, since this did not alter DAT phosphorylation; therefore, PREP-induced decrease in phosphorylation of DAT is independent of ERK. Additionally, the differences in Wt and PREPko cells on DA uptake became visible only in the same setup. It is possible that the stress caused by transfection of several constructs enhances PREP functions in Wt HEK-293 cells. PREP is a stress-sensitive enzyme [45], and it has been connected with stress-induced cell death [46]; we have also shown that PREP inhibition has a more prone effect, e.g., on autophagy under stress [29]. Therefore, it is possible that different stress levels caused by transfections alter the physiological functions of PREP. Additionally, a more significant impact of PREP deletion than PREP inhibition on DAT phosphorylation suggests a non-hydrolytic regulation of PREP protein on DAT. This was also supported by the fact that inactive PREP caused changes, e.g., in ERK phosphorylation when restored to PREPko cells.

Activation of PKC has been shown to phosphorylate DAT directly on its PKC domain, but active PKC can also phosphorylate the DAT MAPK/SH3 domain via ERK [3,5,41]. Phosphorylation of the MAPK/SH3 domain upregulates and phosphorylation of the PKC domain downregulates the membrane localization of DAT. Earlier reports show that PKC activation leads to decreased DAT on the plasma membrane, indicating that phosphorylation of the PKC domain has a primary role in regulation of DAT localization and function [5,9,47]. In the current study, the PKC activator PMA elevated the phosphorylation of DAT and ERK, and caused a minor increase in DAT oligomerization and a decrease in DA uptake, as expected. The pDAT antibody used in the study is targeted for phosphorylation of Thr53 [40], which should bind DAT more to the plasma membrane, but phosphorylation of both domains by PKC activation could explain our results. Although KYP-2047 did not have a direct effect on phosphorylation of DAT and ERK, it seemed to potentiate some of the effects of PMA and Gö-6983. KYP-2047 had a trend toward a further decrease in pDAT levels when combined with Gö-6983, but then again, PREP inhibition elevated oDAT with PMA, while it decreased them when combined with Gö-6983. Particularly, the potentiating effect of KYP-2047 on PMA showed significant effects; the DA uptake assay KYP-2047+PMA significantly decreased DA uptake in cells, and KYP-2047 elevated the PMA-induced PKC activity. Increased PKC activity and elevated oDAT combined with pDAT can explain the functional changes of DAT, and support the intracellular nature of oDAT. KYP-2047 alone showed the opposite effect on oDAT levels in the PMA and Gö-6983 groups. It is possible that the starvation before PMA treatment can change the effect of KYP-2047 treatment. The mechanism behind this warrants further studies, but PKC activation has been reported to inhibit starvation-induced autophagy [48], and PREP inhibition can increase autophagy [29]. Therefore, it is possible that starvation induces changes in the PREP-regulated network, which changes the impact of a PREP inhibitor.

Since PREP has also been shown to increase the aggregation and toxicity of aSyn [49], and aSyn binds to DAT and regulates DAT function by stabilizing it on the plasma membrane under normal conditions [17,18,50], we studied the interplay between these proteins on DAT functions. Overexpression of aSyn can elevate DAT on the plasma membrane partly by stabilizing it on the membrane, and partly since aSyn and DAT interaction also decreases phosphorylation of DAT [17,18], causing the accelerated uptake of DA and high intracellular DA concentration, resulting in increased oxidative stress and apoptosis [17]. Elevated DA can also directly increase aSyn toxicity by increasing the formation of toxic aSyn oligomers [19,20]. In our study, overexpression of aSyn alone did not have a statistically significant effect on the pDAT and oDAT, or on the pERK/ERK ratio in Wt or PREPko HEK-293 cells. aSyn overexpression decreased pDAT levels in PREPko cells, indicating that elevated aSyn binds more DAT on the cell membrane, but no effect was seen in oligomeric DAT, and the changes were not significant. A combination of DAT and PREP caused a significant decrease in DA uptake, particularly in PREPko cells, but this combination caused variable but not significant changes in pDAT or oDAT, or in the pERK/ERK ratio. It is possible that overexpression of various proteins, particularly aSyn, may block the transport of DAT on the cellular membrane as suggested by [51], although its phosphorylation was reduced. Similar to Wt and PREPko cells, PREP inhibition did not cause changes in pDAT or DAT oligomerization in this setup, but significantly increased ERK phosphorylation in all groups. This was in contrast to starved and non-starved cells, but it is possible that as a stress-sensitive kinase, ERK reacts to transfection of various proteins in cells, and the same may also occur with PREP, as discussed above.

In conclusion, our results confirmed the previous observations that PREP regulates DAT function by regulating phosphorylation of DAT. The results showed that PREP inhibitors regulated phosphorylation of ERK, but the mechanism in regulation of DAT phosphorylation was not dependent on PKC activation or phosphorylation of ERK. Additionally, PREP appeared to regulate DAT oligomerization, either via phosphorylation or by transport mechanisms.

## 4. Materials and Methods

### 4.1. Materials

The [^3^H]-DA was obtained from Perkin Elmer (#NET673001MC, Waltham, MA, USA). The reagents were purchased from Sigma-Aldrich (MO, USA) if not otherwise specified. The ethanol was purchased from Altia (Finland). The PREP inhibitor, KYP-2047 (4-phenylbutanoyl-l-prolyl-2(S)-cyanopyrrolidine), was synthesized in the School of Pharmacy (synthesized 2016; stability checked by NMR 2018 and 2020), University of Eastern Finland, as previously described in [52]. The stock solution of KYP-2047 was 100 mM in dimethyl sulfoxide (DMSO) and was diluted with PBS to 1 mM concentration prior to dilution to the final concentration in the cell medium. Plasmid construction was performed as described in [29].

### 4.2. Cell Lines

Cell cultures were maintained at 37 °C in 5% CO_2_/humidified air, and passage numbers 10–30 were used in the assays. Cells were grown to confluency in 75- and 175-cm^2^ flasks and split twice weekly.

Human embryonic kidney 293 (HEK-293) cells (ATCC, Manassas, VA, USA, RRID: CVCL_0045) and stable PREP knock-out (PREPko) cells generated in HEK-293 background were used in the study. HEK-293 cell line is listed as a commonly misidentified cell line by the International Cell Line Authentication Committee (ICLAC; http://iclac.org/databases/cross-contaminations/; version 10; released 25 March 2020). The HEK-293 cell line was authenticated by the Genomics Unit of Technology Centre, Institute for Molecular Medicine Finland (FIMM) with the Promega GenePrint24 System after finishing the experiments in October 2018.

PREPko cells were generated as described in [49]. Shortly, PREPko cells were generated using CRISPR-cas9n plasmid (pSpCas9n(BB)-2A-Puro (PX462) V2.0; Addgene plasmid; #62987) targeted at the 3rd exon of the PREP gene60. Two oligonucleotides for CRISPR guide A (5′atggcacagtaatctt) and B (5′cttgagcagtgtccca) were designed and annealed separately. The backbone PX462 was digested with BbsI restriction enzyme (#R0539S, New England Biolabs, Ipswich, MA, USA) and ligated with Ligate-IT (USB HEK-293 ce™ Rapid Ligation Kit, Affymetrix, Santa Clara, CA, USA).

Wild-type (Wt) HEK-293 cells were grown in Dulbecco´s modified Eagle´s medium (DMEM) containing an additional 10% (*v*/*v*) fetal bovine serum (FBS; Invitrogen, Carlsbad, CA, USA) and 1% (*v*/*v*) penicillin-streptomycin (Lonza, Basel, Switzerland). PREPko cells were cultured in DMEM containing an additional 20% (*v*/*v*) FBS (Invtrogen) and 1% (*v*/*v*) penicillin-streptomycin (Lonza). Cell cultures were maintained at +37 °C in 5% CO2 in a humidified incubator, and used in passages 3–18 for the experiments.

### 4.3. Plasmids

The pAAV1-EF1α-hPREP (#59967; Addgene, Cambridge, MA, USA), pAAV1-EF1α-S554A-hPREP (S554A-PREP; #59968; Addgene), and pAAV EF1a V5-aSyn(WT) (#60057; Addgene) have been described previously [29]. The pAAV-EF1a control vector with 50 bp insert was created by annealing complementary oligonucleotides (10× annealing buffer: 100 nM Tris HCl, 500 mM NaCl, 10 mM EDTA). This insert was recombined into the KpnI-HF (R3142; NEB, Ipswich, USA) and EcoRV-HF (R3195; NEB) sites of pAAV-EF1a-PREP vector using an In-Fusion HD cloning kit (639645; Clontech, Mountain View, CA, USA). Plasmid was transformed into Stbl3 One-Shot competent cells (#C737303, Thermo Fisher Scientific, Waltham, MA, USA). An insert containing the clone was verified by sequencing. The pcDNA3.1-hDAT (#32810; Addgene) was a gift from Dr. Susan Amara.

### 4.4. Western Blotting

Wt HEK-293 and PREPko cells were seeded (6 × 105 cells per well) the day before transfection on 6-well plates. Cells were transfected with Lipofectamine 3000 transfection kit (Thermo Fisher) according to manufacturer´s protocol 2 days prior to experiment. To study ERK phosphorylation, DAT-transfected cells were starved for 2 h in serum-free media prior to treatment with vehicle (0.01% DMSO (*v*/*v*)), 1 µM KYP-2047, 1 µM phorbol 12-myristate 13-acetate (PMA), or 1 µM PMA + 1 µM KYP-2047 for 30 min. DAT-transfected cells were treated with vehicle (0.01% DMSO (*v*/*v*)), 1 µM KYP-2047, 1 µM Gö-6983, or 1 µM Gö-6983 + 1 µM KYP-2047 for 30 min without starvation.

Cells were lysed in 150 µl ice-cold modified RIPA buffer (50 mM Tris HCl pH 7.4, 1% NP-40, 0.25% sodium deoxycholate, 150 mM NaCl) containing Halt Phosphatase Inhibitor (Product# 87786, Thermo Fisher Scientific, IL, USA; purchased 2019) and Halt Protease Inhibitor cocktail (Product# 78430, Thermo Fisher Scientific; purchased 2019). Cells were homogenized with an ultrasound sonicator (GM35-400, Rinco Ultrasonic, Switzerland). After centrifugation (13,300× *g*, +4 °C, 15 min), the supernatants were collected. Protein concentration was measured by the bicinchoninic acid (BCA) method (Product #23225, Thermo Fisher Scientific). For SDS-PAGE, 30 μg of protein was loaded onto a 12% Mini-PROTEAN TGX precast gel (Product # 4561044, Bio-Rad, CA, USA; purchased 2018–2020). Proteins were transferred to PVDF membrane (Trans-blot Turbo, Product# 1620175, Bio-Rad) by using a Trans-blot Turbo Transfer System (Bio-Rad). The membranes were blocked with 5% skim milk in Tris-buffered saline containing 0.1% Tween-20 (TTBS). The membranes were incubated with a primary antibody overnight at +4 °C and thereafter with a secondary antibody (2 h, RT). Antibodies were diluted in 5% skim milk in TTBS. The following primary antibodies and dilutions were used: DAT, rat anti-DAT (#MAB369, Merck Millipore, Darmstadt, Germany, dilution 1:1000, RRID: AB_2190413); rabbit anti-phospho (T53)-DAT (pDAT) (#ab183486, AbCam, UK, dilution 1:500, RRID: AB_2756345); ERK, rabbit anti-ERK1+ERK2 (#ab17942, AbCam, UK, dilution 1:1000, RRID: AB_2297336); phospho ERK (pERK), mouse anti-ERK1+ERK2 (#ab50011, Abcam, dilution 1:5000, RRID: AB_1603684); β-actin, rabbit anti-β-actin (#4967S, Signaling Technology, MA, USA, dilution 1:1000, RRID: AB_330288). Goat anti-rabbit HRP (#31460, Thermo Fisher Scientific, RRID: AB_228341), goat anti-rat HRP (#ab97057, AbCam, RID: AB_10680316), and goat anti-mouse HRP (#31430, Thermo Fisher Scientific, RRID: AB_228307) were used for secondary antibodies (dilution 1:2000). Validation data for the DAT and pDAT antibodies are provided in Appendix A. Our validation data shows that the MAB369 DAT antibody was specific for oligomeric DAT (oDAT) in the current study setup.

The images were captured with a ChemiDoc MP chemiluminescence scanner (Bio-Rad). Optical-density (OD) values of the WB images were analyzed using ImageJ software (NIH, MD, USA). The OD values were normalized to the loading control (β-actin) ODs. pERK/ERK ratio was calculated from values normalized to β-actin. Wt control was set as 100%. Data are from 6 or 8 independent WB experiments, and samples for WB were collected from 6 or 8 independent cell culture preparations. Raw Western blot images and individual values of analysis are shown in Appendix A.

### 4.5. PKC Activity Assay

PKC activity was measured from HEK-293 cells with a PKC Kinase Activity Assay Kit (ab139437, Abcam; purchased 2019) according to the manufacturer’s instructions. Briefly, cells were plated on 12-well plates coated with Poly-L-lysine (Sigma) (250,000 cells per well). The following day, the cells were treated with vehicle (0.1% DMSO), KYP (10 µM), PMA (100 nM, Sigma), Gö-6983 (1 µM), KYP (10 µM) together with PMA (100 nM), or Gö-6983 and PMA (100 nM). When Gö-6983 was used in combination with PMA, it was added to the cells 10 min before PMA. The cells were treated for 30 min at 37 °C, and the treatments were done in DMEM without FBS. After the treatment, the cells were washed once with PBS and lysed with lysis buffer containing 50 mM KH2PO4, 1.5 mM MgCl2, 10 mM NaCl, and 1 mM EDTA (pH 7.4) on ice. The cells were scraped, collected to Eppendorf tubes, and centrifuged (18,000× *g*, 15 min, +4 °C). Supernatants were collected and diluted 1:20 to the kinase assay dilution buffer provided in the kit. These diluted samples were pipetted to the 96-well plate included in the kit, and the following steps were done according to the manufacturer’s instructions. The protein concentrations of the samples were measured with BCA to normalize the PKC activities to the protein levels. Data are from 3 independent experiments for cells with 2 biological replicates.

### 4.6. ^3^H-Dopamine Uptake Assay

Optiphase HiSafe3 (Perkin Elmer, Turku, Finland) scintillant was added to the wells, and radioactivity was quantified using a scintillation counter (Wallac 1450 MicroBeta TriLux Liquid Scintillation Counter and Luminometer, Turku, Finland).

Cells were seeded (1.2 × 105 cells per well or 1.5 × 105 cells per well) the day before transfection in 24-well plates (Falcon 3047) coated with poly-l-lysine (Sigma-Aldrich). Cells were transfected 2 days before the experiment with hDAT or hDAT in combination with mock plasmid, PREP or S554A-PREP using Lipofectamine 3000 (Thermo Fisher) transfection protocol. The medium was removed, and cells were washed once with 0.25 mL uptake buffer (5 mM Tris, 7.5 mM HEPES, 120 mM NaCl, 5.4 mM KCl, 1.2 mM CaCl2, 1.2 mM MgSO4, 1 mM ascorbic acid, 5 mM glucose; pH 7.1). Cells were preincubated in uptake buffer for 15 min or in uptake buffer containing 1 µM KYP-2047, 1 µM PMA, 1 µM Gö-6983, 1 µM PMA + 1 µM KYP-2047, or 1 µM Gö-6983 + 1 µM KYP-2047 for 30 min at +37 °C prior to the addition of 0.25 mL 20 nM [^3^H] DA with 2 µM or 10 µM unlabeled DA (PerkinElmer, Woodbridge, ON, Canada) and incubation for 5 min at +37 °C. Cells treated with Gö-6983 were preincubated in media containing 1 µM Gö-6983 for 10 min before other treatments. PMA-treated cells and their control cells were starved in serum-free media for 2 h before treatments. Wells were rinsed three times with ice-cold 0.32 M sucrose and lysed with 0.25 mL of 0.25 M NaOH. Optiphase SuperMix (Perkin Elmer, Turku, Finland) scintillant was added to the wells, and radioactivity was quantified using a scintillation counter (Wallac 1450 MicroBeta TriLux Liquid Scintillation Counter and Luminometer, Turku, Finland) after placing crosstalk-minimizing inserts (1450–109, Perkin Elmer) into each well. Counts per well were measured for 5 min. Data are from 3 independent experiments for cells with 3 biological replicates, and individual values of the uptake assay are shown in Appendix A.

### 4.7. Statistical Analysis

Statistical analyses were performed using either GraphPad Prism (version 6.02, GraphPad Software, Inc., San Diego, CA, USA) or SPSS Statistics (Version 22.0.0.1, IBM Corporation, Armonk, NY, USA) software. One-way ANOVA and two-way ANOVA with Tukey´s post-hoc test were used as statistical tests. Normality of the data was evaluated with a Shapiro–Wilk test. Data are presented as mean ± standard error of the mean (SEM). n = number of biological replicates from independent cell culture preparations in the figures legends. The values above mean+2*standard deviation (SD) or below mean-2*SD were defined as outliers and were removed from the data analysis. The exact *p*-values are provided in the results section. The results were considered statistically significant at *p* < 0.05.

## Figures and Tables

**Figure 1 ijms-22-01777-f001:**
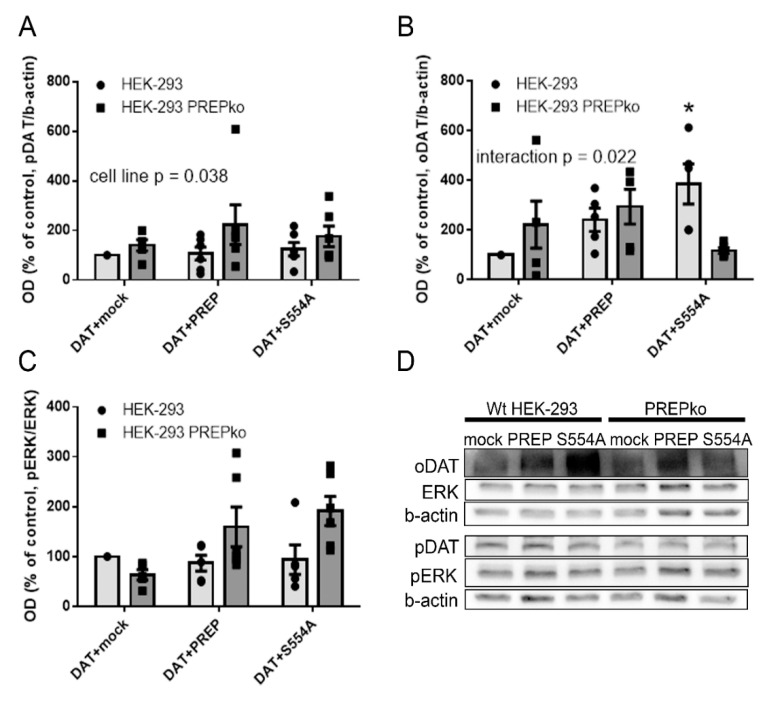
Prolyl oligopeptidase (PREP) regulates phosphorylation and oligomerization of dopamine transporter (DAT), and phosphorylation of extracellular signal-regulated kinase (ERK). Oligomeric DAT (oDAT), phosphorylated DAT (pDAT), ERK, and phosphorylated (pERK) were measured by Western blot (WB) in Wt HEK-293 and stable PREPko HEK-293 cells transfected with DAT and mock, DAT and PREP, or DAT and inactive PREP (S554A-PREP). Optical-density values were normalized to the loading control (β-actin) optical density. Representative WB figures for (**A**–**C**) are presented in panel (**D**). Data were collected from six independent WBs that were done with samples from six independent cell culture preparations. Bars represent mean ± SEM., * *p* = 0.041 one-way ANOVA, Tukey´s post-hoc test. Other *p*-values are from two-way ANOVA.

**Figure 2 ijms-22-01777-f002:**
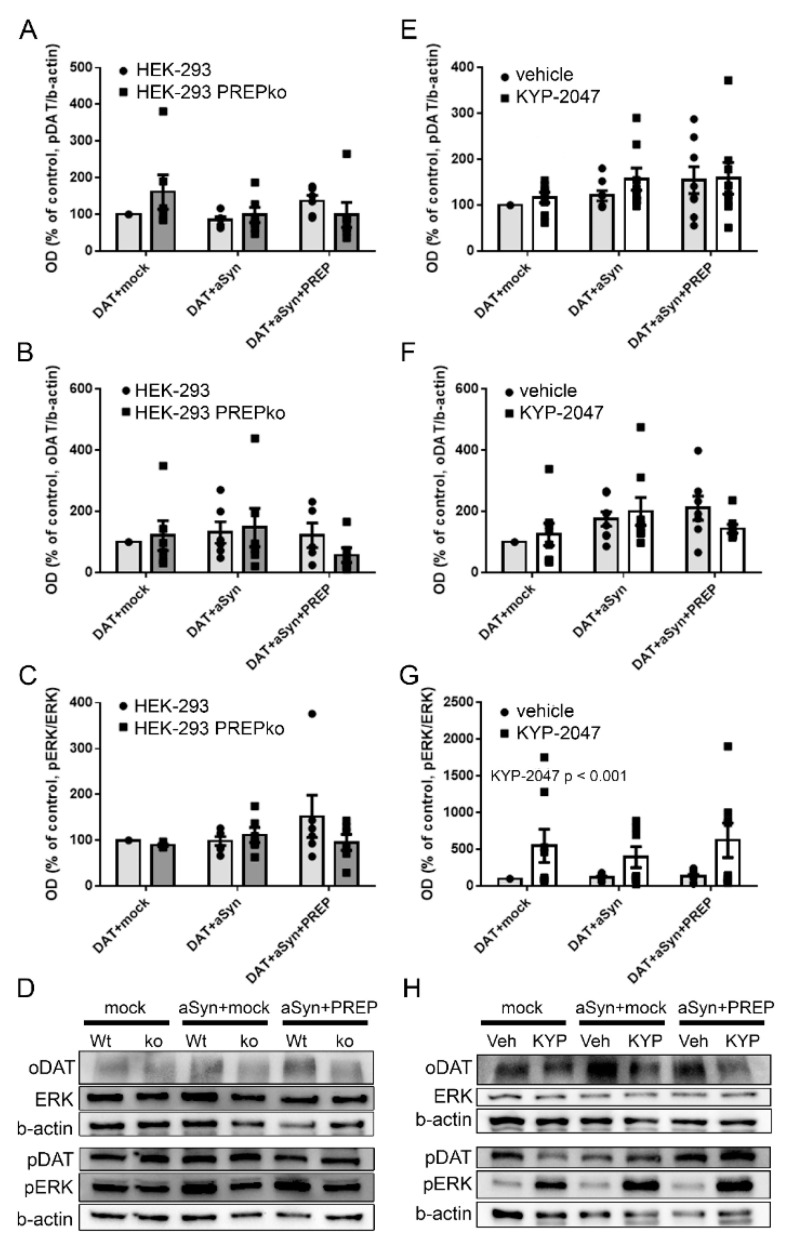
PREP inhibition elevates phosphorylation of ERK but does not have an effect on phosphorylation of DAT. Wt HEK-293 cells and PREPko HEK-293 cells were transfected with DAT and mock, DAT and aSyn or DAT, aSyn, and PREP (**A**–**D**). Wt HEK-293 cells with similar transfections were treated with vehicle or 1 µM KYP for 30 min prior to cell lysis 48 h after transfection (**E**–**H**). The oDAT, pDAT, ERK, and pERK were measured by WB. Optical-density values were normalized to the loading control (β-actin) optical density. Overexpression of aSyn or a combination of aSyn and PREP did not have a statistically significant effect on phosphorylation (**A**) or oligomerization of DAT (**B**), and deletion or inhibition of PREP did not change the effect. However, PREP inhibition increased phosphorylation of ERK (**G**), but PREP deletion did not (**C**). Representative WB for figures (**A**–**C**,**G**), and for (**E**–**G**,**H**). Data were collected from 6–8 independent WBs that were done with samples from 6–8 independent cell culture preparations. Bars represent mean ± SEM. *p*-value is from two-way ANOVA.

**Figure 3 ijms-22-01777-f003:**
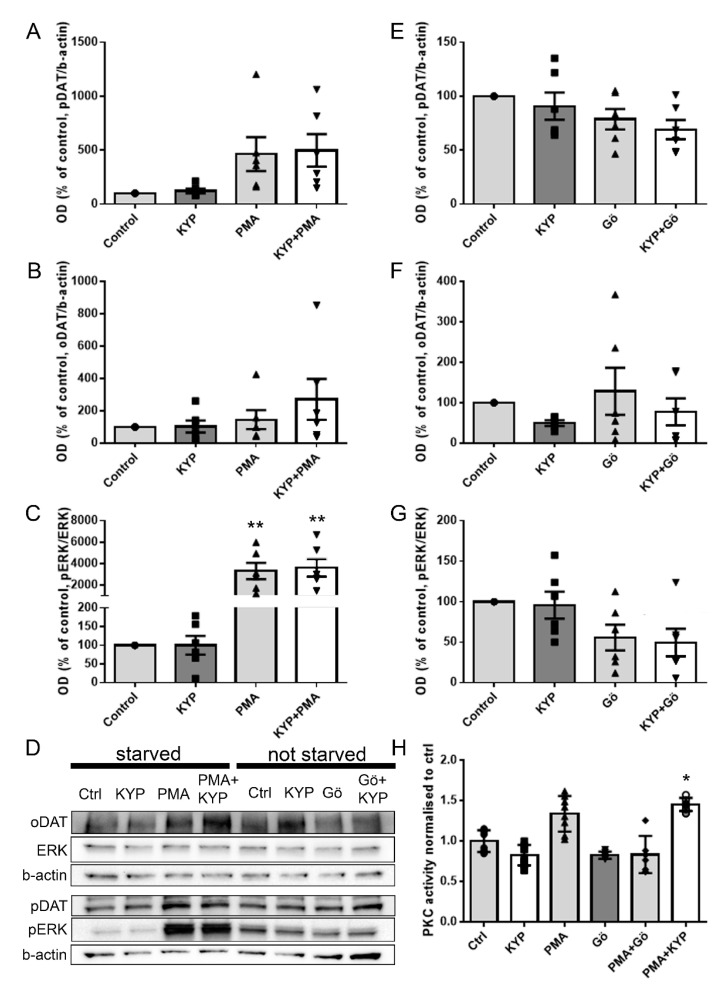
The effect of PREP inhibitor KYP-2047 alone or in combination with PKC activator PMA or inhibitor Gö-6983 on phosphorylation of DAT and ERK, oligomerization of DAT, and PKC activity in HEK-293 cells. DAT, pDAT, ERK, and pERK were measured by WB in Wt HEK-293 cells transfected with DAT. Cells were starved in serum-free media for 2 h before 1 µM KYP, 1 µM PMA, or 1 µM KYP + 1 µM PMA treatment for 30 min (**A**–**C**); or treated with 1 µM KYP, 1 µM Gö-9683, or 1 µM KYP + 1 µM Gö-6983 for 30 min without starving (**E**–**G**). Optical-density values were normalized to the loading control (β-actin) optical density. Representative WB figures for (**A**–**C**) and (**D**–**F**) are presented in panel (**G**). PKC activity assay was performed on HEK-293 cells treated with 0.1% DMSO (Ctrl), 10 µM KYP, 100 nM PMA, 1 µM Gö-6983, KYP 10 µM KYP + 100 nM PMA, or 1 µM Gö-6983 + 100 nM PMA for 30 min in FBS-free medium (**H**). Squares, triangles and circles in the Figure present individual data points. Data were collected from six WBs that had samples from six independent cell culture preparations. PKC activity assay data are from three independent experiments for cells with two biological replicates. Bars represent mean ± SEM. * *p* < 0.05, ** *p* < 0.01, one-way ANOVA, Tukey´s post-hoc test.

**Figure 4 ijms-22-01777-f004:**
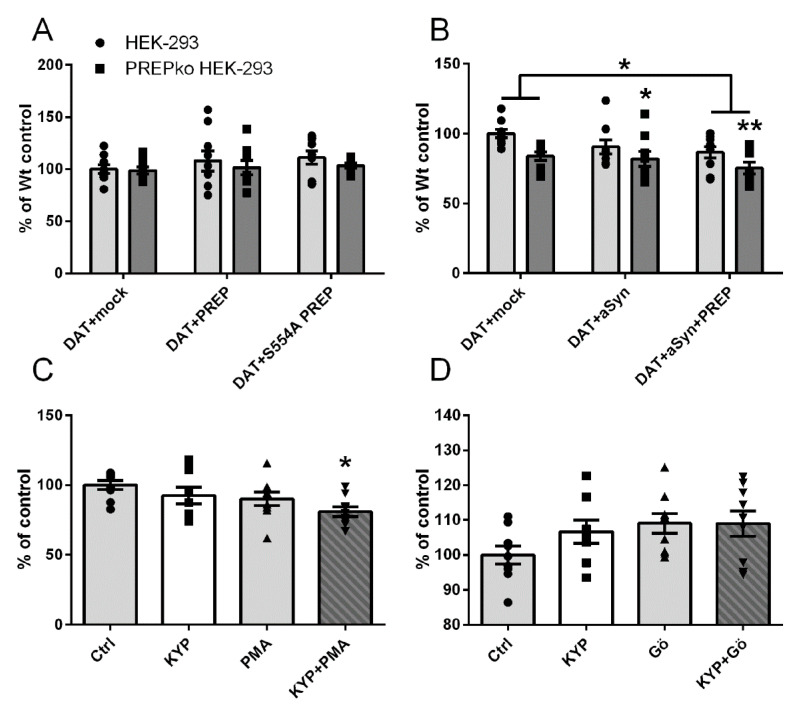
The effect of PREP on dopamine uptake in Wt and stable PREPko cells. Uptake of 3H-dopamine was measured in Wt and stable PREPko cells transfected with DAT and active or inactive PREP (**A**), or DAT and α-synuclein (aSyn) with or without PREP (**B**), and in DAT-transfected Wt cells that were treated 30 min with PREP inhibitor and PKC activator (PMA) or inhibitor (Gö-6983) (**C**,**D**). Cells were starved in serum-free media 2 h before treatment and then treated with 1 µM KYP, 1 µM PMA, or 1 µM KYP + 1 µM PMA for 30 min (**C**). Cells treated with 1 µM KYP, 1 µM Gö-9683, and 1 µM KYP + 1 µM Gö-6983 for 30 min were not starved (**D**). Data were from three independent experiments for cells with three biological replicates. Bars represent mean ± SEM. * *p* < 0.05, ** *p* < 0.01, one-way ANOVA or two-way ANOVA, Tukey´s post-hoc test.

## Data Availability

All data is available from the corresponding author by request.

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
