# Peer review of "Prolyl Oligopeptidase Regulates Dopamine Transporter Oligomerization and Phosphorylation in a PKC- and ERK-Independent Manner"

_ijms, 2021, doi:10.3390/ijms22041777_

Round 1

Reviewer 1 Report

This manuscript is well written, and the data and their statistical analyses were solid. The conclusions made were well supported by several independent experimental approaches.

Previous studies from this lab showed that a serine protease, prolyl oligopeptidase (PREP) regulates DAT phosphorylation and internalization in the mouse striatum. Here, in this study they confirmed their previous findings that lack of PREP can increase phosphorylation and internalization of DAT and decrease uptake of dopamine. Overall this is a well-executed study characterizing the mechanisms of PREP on regulating DAT function and demonstrated that PREP-mediated phosphorylation, oligomerization, and internalization of DAT is not dependent on PKC or ERK.This could have implications for novel targeting strategies at these transporters to modulate their activity. Thus, the findings are important and should be of interest to readers. I have only minor comments that are mostly meant to improve the readability of the manuscript.

I do not see any criticism that would require additional work or textual changes.

Author Response

We wish to thank the referee for very positive comments on our manuscript.

Reviewer 2 Report

In the manuscript entitled “Prolyl oligopeptidase regulates dopamine transporter oli
gomerization and phosphorylation in a PKC and ERK inde- 3
pendent manner” written by Ulrika H. Julku and collaborators have discovered that prolyl oligopeptidase-mediated phosphorylation, oligomerization and internalization of dopamine transporter is not dependent on protein kinase C or extracellular signal-regulated kinase.

This manuscript is well written and provides the information of stated results.

Nevertheless, I would suggest the following recommendations for the authors to address and strengthen their manuscript.

Abstract is comprehensive and cover all the requisite aspect of the issue.

I would ask for adding the methods (a few words) used by which the results are achieved. Lines 17-20 ‘In this study, we clarified the mechanism behind this by using HEK-293 17and PREP knock-out HEK-293 cells with DAT transfection. Additionally, we tested the effects of 18PREP inhibition and alpha-synuclein on PREP-related DAT regulation, and characterized impact of 19PREP on protein kinase C (PKC) and extracellular signal-regulated kinase (ERK) as these kinases 20regulate DAT phosphorylation.’

The Introduction part is well elaborated. I have only two suggestions for this part: line 34-35 For the sentence ‘Dopaminergic neurotransmission is impaired in many diseases such as Parkinson´s disease, depression, bipolar disorder, attention deficit hyperactivity disorder, and addictions [1,2]’. Schizophrenia could be added as the disease  that  is also characterized by the impairment of  dopaminergic neurotransmission. The last sentence (lines 80-82) ‘In this study we aimed to characterize PREP as a regulator of DAT and tried to find the mechanism explaining the previous findings with PREP, DAT, and ERK.’ could included methods used to acieved this aim.

In the Result part Figure 2 (A-G) and Figure S5: does the control group reprezent only one dot n=1? There is only one dot visualized.

The Discussion part is well written and clear.

Author Response

We wish to thank the reviewer for positive and constructive comments on our manuscript.

  • I would ask for adding the methods (a few words) used by which the results are achieved.
  • A: We have now modified this part of abstract as "In this study, we clarified the mechanism behind this by using HEK-293 and PREP knock-out HEK-293 cells with DAT transfection. We tested the effects of PREP, PREP inhibition and alpha-synuclein on PREP-related DAT regulation by using Western blot and dopamine uptake assay, and characterized impact of PREP on protein kinase C (PKC) and extracellular signal-regulated kinase (ERK) by using PKC assay and Western blot, respectively, as these kinases regulate DAT phosphorylation." We hope this is now acceptable.

  •  I have only two suggestions for this part: line 34-35 For the sentence ‘Dopaminergic neurotransmission is impaired in many diseases such as Parkinson´s disease, depression, bipolar disorder, attention deficit hyperactivity disorder, and addictions [1,2]’. Schizophrenia could be added as the disease  that  is also characterized by the impairment of  dopaminergic neurotransmission. The last sentence (lines 80-82) ‘In this study we aimed to characterize PREP as a regulator of DAT and tried to find the mechanism explaining the previous findings with PREP, DAT, and ERK.’ could included methods used to achieved this aim.
  • A: We have now added schizophrenia as one of the diseases where dopamine has a role. Additionally, the last sentence in the introduction is now modified as follows: " In this study, we aimed to characterize PREP as a regulator of DAT and tried to find the mechanism explaining the previous findings with PREP, DAT, and ERK by using HEK-293 cells transfected with DAT, Western blot and dopamine uptake assay in presence and absence of PREP. Additionally, we characterized the impact of aSyn and PREP inhibition on DAT, and evaluated if PREP-mediated effect is based on PKC or ERK regulation by using kinase assay and Western blot. " We hope this is now acceptable.

  • In the Result part Figure 2 (A-G) and Figure S5: does the control group reprezent only one dot n=1? There is only one dot visualized.
  • A: The reviewer has raised an important question. In the Western blot figures, the wild-type control was set to 100% to analyze the effect of different treatments. In all Western blots, we had one wild-type control to have room for all treatments, and therefore, the average and dots in graphs show only 100% value. We have now highlighted this sentence in the Materials and Methods, Western blot "Wt control was set as 100 %. Data are from 6 or 8 independent WB experiments, and samples for WB were collected from 6 or 8 independent cell culture preparations. " We hope we have answered to reviewer's question.